# Approximation Algorithms for Observer Aware MDPs

**Shuwa Miura**[1]        **Olivier Buffet**[2]        **Shlomo Zilberstein**[1]

[1]University of Massachusetts, Amherst, MA, USA
[2]Université de Lorraine, INRIA, CNRS, LORIA, Nancy, France

## Abstract

We present approximation algorithms for Observer-Aware Markov Decision Processes (OAMDPs). OAMDPs model sequential decision-making problems in which rewards depend on the beliefs of an observer about the goals, intentions, or capabilities of the observed agent. The first proposed algorithm is a grid-based value iteration (Grid-VI), which discretizes the observer's belief into regular grids. Based on the same discretization, the second proposed algorithm is a variant of Real-Time Dynamic Programming (RTDP) called Grid-RTDP. Unlike Grid-VI, Grid-RTDP focuses its updates on promising states using heuristic estimates. We provide theoretical guarantees of the proposed algorithms and demonstrate that Grid-RTDP has a good anytime performance comparable to the existing approach without performance guarantees.

## 1 INTRODUCTION

Effective communication of intentions, goals, and desires is crucial in our daily interactions and is equally vital for autonomous agents. For instance, consider an autonomous vehicle (AV) approaching a crosswalk with a pedestrian nearby. The AV might approach the crosswalk at high speed and then decelerate just in time to avoid hitting the pedestrian. However, this can be unsettling for the pedestrian. A more reassuring approach would be for the AV to slow down well before reaching the crosswalk, signaling its intention to stop. We term such actions that take into account the perspective or beliefs of an observing agent as *observer-aware* behaviors. Observer-aware behaviors include making the agent's goal clear [Dragan and Srinivasa, 2013], demonstrating its capabilities [Kwon et al., 2018] or disguising possible intentions [Masters and Sardina, 2017, Savas et al., 2022].

The Observer-Aware Markov Decision Process (OAMDP)

[Miura and Zilberstein, 2021] offers a general framework for producing observer-aware behaviors. The OAMDP framework assumes a model of how the agent's actions would be interpreted by the observer. In OAMDPs, possible goals, intentions, or capabilities of the observed agent are represented as types. After the observed agent takes an action, the observing agent updates its belief over the possible types, which determines the reward function.

While OAMDP allows modeling various observer-aware planning problems in a unified way, solving OAMDPs has been shown to be intractable in the worst case [Miura and Zilberstein, 2021]. The intractability stems from the fact that rewards depend on the belief of the observer, which in turn depends on the history so far. Previous work proposed using Monte-Carlo Tree Search (MCTS) to solve OAMDPs for the finite-horizon objective [Miura and Zilberstein, 2021]. While MCTS exhibits good anytime behavior, it does not provide guarantees on the qualities of the resulting policies.

In this paper, we propose the first approximation algorithms for OAMDPs. We begin by establishing that the domain state and the observer's belief are sufficient statistics in OAMDPs (Proposition 1). Our first proposed algorithm is a grid-based value iteration (Grid-VI), which discretizes the belief of the observer into regular grids. We show that Grid-VI converges to the unique fixpoint both in discounted (Proposition 4) and undiscounted (Proposition 6) settings under the standard assumptions, and provide error bounds for the discounted setting (Proposition 5). A potential drawback of Grid-VI is that it can waste time updating values at irrelevant states. To address the issue, we propose a variant of Real-Time Dynamic Programming (RTDP) [Barto et al., 1995] to solve OAMDPs, called Grid-RTDP. Grid-RTDP utilizes heuristic estimates to focus updates on promising states. We show that Grid-RTDP retains a key desirable property of RTDP (Proposition 7). Our experimental results indicate that our proposed algorithms are capable of computing near-optimal policies. Specifically, Grid-RTDP solves problems significantly faster than Grid-VI and offers anytime performance comparable to MCTS.

## 2 BACKGROUNDS AND NOTATIONS

### 2.1 MARKOV DECISION PROCESSES

A finite Markov decision process (MDP) models sequential decision-making under uncertainty. An MDP is described by a tuple $M = \langle S, A, T, R, \gamma, d_0 \rangle$. $S$ and $A$ are finite sets of states and actions, respectively. $S_t$ and $A_t$ represent a state and an action at time $t$. $T(s_t, a_t, s_{t+1})$ is the probability of $S_{t+1}=s_{t+1}$ when $A_t=a_t$ and $S_t=s_t$. $R$ is a reward for taking $a_t$ at $s_t$. $\gamma$ is a parameter called the discount factor. $d_0$ is the initial state distribution $S_0 \sim d_0$.

A solution to an MDP is called a *policy* ($\pi$). We use the following two types of policies in the paper. A *stationary policy* is a conditional distribution of actions given a state. A *history-dependent policy* is a conditional distribution of actions given a history, where a history $h_{t+1}$ is a sequence of state-action pairs up to time $t$ and the last visited state $s_{t+1}$. An optimal policy for an MDP is a policy that maximizes $\mathbb{E}[\sum_{t=0}^{\infty} \gamma^t R(S_t, A_t)|d_0, \pi]$. A policy ($\pi$) induces a value function $V^\pi(s) = \mathbb{E}[\gamma^t R(S_t, A_t)|S_0 = s, \pi]$. The optimal value function $V^*$ is the value function corresponding to an optimal policy.

### 2.2 STOCHASTIC SHORTEST PATH PROBLEMS

A *stochastic shortest path problem* (SSP) is an undiscounted, cost-based counterpart of an MDP. An SSP is represented by a tuple $\langle S, A, T, C, d_0, G \rangle$ where: $S$, $A$, and $T$ are the same as in an MDP. $C(s_t, a_t) : S \times A \to \mathbb{R}_+$ is the cost of performing $a_t$ at $s_t$. $d_0$ is the initial state distribution. $G \subset S$ is a set of goal states. The goal states are absorbing, and transitions out of goal states have zero costs.

A solution of an SSP is a *policy*. An *optimal policy* $\pi^*$ is a policy that minimizes $\mathbb{E}[\sum_{t=0}^{\infty} C(S_t, A_t)|d_0, \pi]$. We restrict our attention to problems in which there exists at least one *proper policy*, which reaches the goal from all states with probability 1, and any improper policies incur infinite costs. Under this assumption, an SSP is guaranteed to have an optimal policy that is proper [Bertsekas and Tsitsiklis, 1991].

### 2.3 OBSERVER-AWARE MDPS

Observer-Aware Markov Decision Processes (OAMDPs) extend MDPs by allowing the reward to depend on the observer's assumed belief over the types of the observed agent [Miura and Zilberstein, 2021].

**Definition.** An OAMDP is a tuple[1]

[1]The original work [Miura and Zilberstein, 2021] allowed an arbitrary function from $H^*$ to $\Delta^{|\Theta|}$ to update the observer's belief. Here, we restrict our attention to a case where the observer updates its belief in a Bayesian fashion.

$M = \langle S, A, T, \gamma, d_0, \Theta, b_0, \tau, R \rangle$ where:

- $S$, $A$, $T$, $\gamma$, and $d_0$ are the same as in MDPs. In this paper, we assume $S$ and $A$ are finite.

- $\Theta$ is a (finite) set of *types*, representing characteristics of the agent, such as possible goals, intentions, or capabilities. The types in OAMDPs are analogous to the types in Bayesian game theory [Harsanyi, 1968].

- $b_0 \in \Delta^{|\Theta|}$ is the initial belief of the observer over the types, where $\Delta^{|\Theta|}$ is a simplex on $\Theta$.

- $\tau : S \times A \times S \times \Theta \to [0, 1]$ is the probability of the observer witnessing a transition $\langle s, a, s' \rangle$ given $s$ and $\theta$. $\tau$ can represent different policies and transition functions of the observed agent depending on types.

- $R : S \times A \times \Delta^{|\Theta|} \to \mathbb{R}$ is a belief-dependent reward function. In this paper, we assume that the rewards can be represented as a linear combination of *domain* and *belief-dependent* rewards. That is, $R(s, a, b)=w_d R_d(s, a) + w_b R_b(b)$ for $w_d, w_b \in \mathbb{R}_+$, where $R_d$ and $R_b$ represent domain and belief-dependent reward, respectively.

After observing a transition $\langle s, a, s' \rangle$, the observer is assumed to be Bayesian rational and updates its belief ($b_t$) using Bayes' rule:

$$b_{t+1}^{s,a,s'}(\theta) = \frac{\tau(a, s'|s, \theta) \cdot b_t(\theta)}{\sum_{\theta' \in \Theta} \tau(a, s'|s, \theta') \cdot b_t(\theta')}. \quad (1)$$

A solution to an OAMDP is a policy that maximizes the expected discounted return:

$$\mathbb{E}[\sum_{t=0}^{\infty} \gamma^t R(S_t, A_t, B_t)|d_0, \pi]. \quad (2)$$

Figure 1 shows an example of an OAMDP with $\Theta = \{\theta_A, \theta_B, \theta_C, \theta_D, \theta_E\}$, where each type corresponds to the observed agent's goal. $\tau(a, s'|s, \theta)$ is typically set to $T^j(s, a, s')\pi_\theta(s, a)$, where $\pi_\theta$ is an assumed policy of the observed agent given a type $\theta$, and $T^j$ is a transition function according to the observer.

Note that since the observer's belief is not directly accessible to the acting agent, the observer's belief in OAMDPs should be understood as a second-order belief. That is, it is a belief the acting agent believes the observer to have.

**Observer's Belief in Approximate Rationality** One possible choice for $\pi_\theta$ is to assume that the acting agent takes an approximately optimal action at each state given their goals, desires, and intentions:

$$\pi_\theta(s, a) \propto \exp^{\beta Q_\theta^*(s,a)}, \quad (3)$$

where $Q_\theta^*$ is the optimal Q-value $Q^*(s, a|\theta) = \mathbf{E}[\sum_{t=0}^{\infty} \gamma^t R_t|S_0=s, A_0=a, \pi^*, \theta]$ representing how good

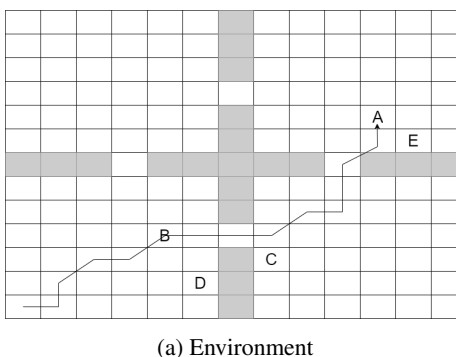

(a) Environment

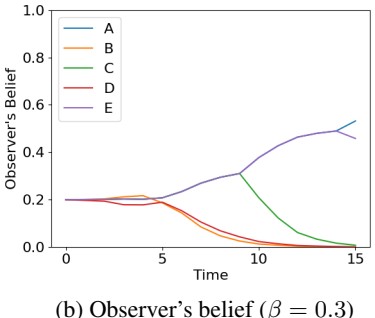

(b) Observer's belief ($\beta = 0.3$)

Figure 1: MazeWorld Domain

$a$ is given $s$ and $\theta$. Note that $Q_\theta^*$ is computed with respect to $M_\theta = \langle S, A, T_\theta, R_\theta, \gamma, d_0 \rangle$ defined for each $\theta \in \Theta$. $\beta \in \mathbb{R}$ serves as a hyperparameter representing the agent's rationality level. Intuitively, it is assumed that the observed agent selects an action with a probability exponentially proportional to the quality of the action. Figure 1b shows the observer's belief changes according to Equation 1 and 3. All the goals are equally likely initially. As the agent moves out of the first room, the beliefs in the goals $B$ and $D$ decrease. By the time the agent enters the top right room, the goals $A$ and $E$ are the two most likely goals.

The Bayesian update (Equation 1) using the Boltzmann action model (Equation 3) is based on the idea that people often infer goals, desires, and intentions from others' behaviors by assuming that their behaviors are approximately rational given their goals, desires, and intentions [Dennett, 1987]. Baker et al. [2009] showed that the Bayesian update using Equation 3 largely agrees with human understanding of goals.

Note that the definition of OAMDPs is not restricted to using the Boltzmann action model as $\pi_\theta$. Other possibilities include assuming that the observed agent follows maximum entropy policies[Ziebart et al., 2008] or boundedly rational policies [Zhi-Xuan et al., 2020] given its type.

**Belief-Dependent Rewards** OAMDP can produce various observer-aware behaviors by changing $R_b$. For instance, to clarify intentions, $R_b$ might be defined as the negative total variation (TV) or the Euclidean distance between the current and target beliefs, where the target belief is $b(\theta^*) = 1$ for the intended type $\theta^* \in \Theta$. On the other hand, if the observed agent wants to obscure its intention, the reward could be the entropy of the observer's belief.

**Relationship to POMDPs and I-POMDPs** While both OAMDPs and partially observable Markov decision processes (POMDPs) operate on the belief of an agent, the two models do not subsume each other. The belief in OAMDP is the second-order belief of the acting agent about the belief of the observer about the type of the acting agent. On the other hand, the belief in POMDP is the first-order belief of

the acting agent about the states of the world. Similarly to how beliefs over states are sufficient for optimal control in POMDPs, we will next show that the current state and the belief of the observer are sufficient statistics in OAMDPs. However, while most solution methods for POMDPs [Monahan, 1982, Pineau et al., 2003] rely on piecewise linear convexity (PWLC) of the value function, the value functions for OAMDPs are not necessarily PWLC. For example, consider using the negative Euclidean distance from the intended type as $R_b$. $R_b$ is not PWLC on $\Delta^{|\Theta|}$. Therefore, solution methods for POMDPs are not directly applicable to OAMDPs.

OAMDPs can be seen as a restricted subset of Interactive POMDPs (IPOMDPs) [Gmytrasiewicz and Prashant, 2005], multi-agent extensions to POMDPs, where agents act by recursively modeling the other agents' beliefs. Several previous works have used IPOMDPs and related multi-agent models to produce observer-aware behaviors[Lo et al., 2020, Alon et al., 2023]. Multi-agent formulations are arguably more general and let us reason about what others do in response based on their beliefs. However, multi-agent formulations are also notoriously hard to solve [Seuken and Zilberstein, 2008]. Miura and Zilberstein [2021] showed that OAMDPs can be seen as a subset of I-POMDPs, where (1) the observer is completely passive, (2) the acting agent knows the observer's type, and (3) the environment is observable to both agents. In the following sections, we will see how our proposed solution methods make use of the additional assumptions.

**Complexity of OAMDPs** While OAMDPs make restrictive assumptions over I-POMDPs, computing an optimal policy has been shown to be PSPACE-hard [Miura and Zilberstein, 2021]. This result suggests that solving OAMDPs is intractable in the worst case. The reduction used in the proof relies on OAMDPs with discontinuous rewards. In this paper, we develop approximation algorithms with provable bounds for OAMDPs with Lipschitz-continuous reward and belief transitions.

## 2.4 OASSPS

In this paper, we also consider OASSPs [Lepers et al., 2024], an undiscounted and cost-based version of OAMDPs. An OASSP is a tuple $\langle S, A, T, d_0, \Theta, b_0, \tau, C, G \rangle$ where $C : S \times A \times \Delta^{|\Theta|} \to \mathbb{R}_+$ is a belief-dependent cost function, and $G$ is a set of goal states. The other components are the same as in OAMDPs. An optimal policy for an OASSP is a policy that minimizes:

$$\mathbb{E}[\sum_{t=0}^{\infty} C(S_t, A_t, B_t) | d_0, \pi]. \tag{4}$$

As in OAMDPs, we assume that $C$ is a linear combination of the domain cost ($C_d$) and belief-dependent cost ($C_b$). That is, $C(s, a, b) = w_d C_d(s, a) + w_b C_b(s, a)$ for $w_d, w_b \in \mathbb{R}_+$. A domain SSP corresponding to an OASSP is an SSP defined as $M_d = \langle S, A, T, d_0, C_d, G \rangle$.

## 3 PROPERTIES OF OAMDPS

In this section, we discuss properties of OAMDPs necessary for developing the proposed algorithms.

### 3.1 SUFFICIENT STATISTICS

To compute policies for OAMDPs, previous work [Miura and Zilberstein, 2021] used a general-purpose method such as UCT [Kocsis and Szepesvári, 2006] to compute history-dependent policies. However, we show that the current state and the belief of the observer contain sufficient information to choose the best action to take:

**Proposition 1.** *The current state and the belief of the observer are sufficient statistics in OAMDPs.*

*Proof.* For all $s_t, s_{t+1} \in S, a_t \in A, b_t \in \Delta^{|\Theta|}, h_t \in H_t$:

$$\Pr(s_{t+1}, b_{t+1} | s_t, a_t, b_t, h_t) \tag{5}$$
$$= \Pr(b_{t+1} | s_t, a_t, s_{t+1}, b_t, h_t) \Pr(s_{t+1} | s_t, a_t, b_t, h_t) \tag{6}$$
$$= [b_{t+1} = b_t^{s_t, a_t, s_{t+1}}] T(s_t, a_t, s_{t+1}) \text{ by definition} \tag{7}$$
$$= \Pr(s_{t+1}, b_{t+1} | s_t, a_t, b_t) \tag{8}$$

where $[\cdot]$ is the Iverson bracket. Moreover, $R$ only depends on $S_t$, $A_t$, and $B_t$ by definition. $\square$

With Proposition 1 in place, we can look for policies of the forms $\pi : S \times \Delta^{|\Theta|} \times A \to [0, 1]$. In other words, we can look for policies to *belief MDP*, whose state space is $S \times \Delta^{|\Theta|}$ instead of $S$. Note that, while the original OAMDP has a finite number of states, the belief MDP has a continuous state space.

## 3.2 DISCONTINUITY IN VALUE FUNCTIONS

Before delving into our proposed algorithms, we address a potential issue in developing a value-based approximation algorithm for OAMDPs. Both of our proposed algorithms approximate values by grouping similar beliefs. This approach operates under the implicit assumption that nearby beliefs should yield similar values. However, we demonstrate that, in a general OAMDP, the rate at which the observer's belief changes can be unbounded, thus invalidating this assumption. To illustrate this issue, consider the following example:

**Example.** *Let us assume that we have an OAMDP with:*

- $\Theta = \{\theta_1, \theta_2, \theta_3\}$,
- $b_1 = (1 - \epsilon, \epsilon, 0) \in \Delta^3$,
- $b_2 = (1 - \epsilon, 0, \epsilon) \in \Delta^3$, *and*
- $\tau^{s,a,s'} = (\tau_0 = 0, \tau_1 > 0, \tau_2 > 0)$.

*Then, $b_1^{s,a,s'} = (0, 1, 0)$ and $b_2^{s,a,s'} = (0, 0, 1)$. Thus,*

$$\frac{\|b_1^{s,a,s'} - b_2^{s,a,s'}\|_\infty}{\|b_1 - b_2\|_\infty} = \frac{\|(0, 1, -1)\|_\infty}{\|(0, \epsilon, -\epsilon)\|_\infty} = \frac{1}{\epsilon}. \tag{9}$$

$\frac{\|b_1^{s,a,s'} - b_2^{s,a,s'}\|_\infty}{\|b_1 - b_2\|_\infty}$ *diverges as $\epsilon \to 0$.*

### 3.3 LIPSCHITZ OAMDPS

Given the potential discontinuity in values, we discuss special cases of OAMDPs with Lipschitz-continuous reward and belief transitions.

**Definition.** An OAMDP is $(L_r, L_p)$-Lipschitz if for all $s, s' \in S, a \in A$, and $b_1, b_2 \in \Delta^{|\Theta|}$:

$$|R(s, a, b) - R(s, a, b')| \le L_r \|b_1 - b_2\|_\infty, \tag{10}$$
$$\|b_1^{s,a,s'} - b_2^{s,a,s'}\|_\infty \le L_p \|b_1 - b_2\|_\infty. \tag{11}$$

Intuitively, in Lipschitz OAMDPs, beliefs close to each other have similar rewards and update to close beliefs. The definition is analogous to Lipschitz continuity of continuous MDPs in general [Rachelson and Lagoudakis, 2010].

Lipschitz continuity of reward and belief transitions can be related to Lipschitz continuity of the value function under a favorable assumption:

**Proposition 2.** *For a $(L_r, L_p)$-Lipschitz OAMDP, if $\gamma L_p < 1$, then $V^*$ is $L_{V^*}$-Lipschitz continuous where:*

$$L_{V^*} = \frac{L_r}{1 - \gamma L_p}. \tag{12}$$

*Proof.* See Appendix A $\square$

As we will see later, Lipschitz continuity of $V^*$ enables us to provide the error bound for discretization (Proposition 5). Note that Proposition 2 states a sufficient condition for Lipschitz continuity of $V^*$. In other words, there could be cases where the conditions of Proposition 2 are not met, but $V^*$ is still Lipschitz. Moreover, in OAMDPs, belief transitions are assumed to be the Bayesian update using Equation 1. We can establish a relationship between the Lipschitz continuity of belief transitions and $\tau$ as follows:

**Proposition 3.** *If $\tau^{s,a,s'}(\theta) > 0$ for all $\theta \in \Theta$, $s, s' \in S$, and $a \in A$, belief transitions are Lipschitz continuous.*

*Proof.* See Appendix A □

For example, using the Boltzmann action model (Equation 3) ensures that $\tau^{s,a,s'}(\theta) > 0$, which guarantees the Lipschitz continuity of belief transitions.

## 4 APPROXIMATION ALGORITHMS

In this section, we propose approximation algorithms for OAMDPs/SSPs. Our first proposed algorithm is a grid-based value iteration (Grid-VI), which discretizes the observer's belief into regular grids. Our second proposed algorithm is a variant of Real-Time Dynamic Programming (RTDP), called Grid-RTDP. Grid-RTDP relies on the same grid-based discretization scheme as Grid-VI, but focuses its updates on promising states using heuristic estimates.

### 4.1 GRID-BASED VALUE ITERATION FOR OAMDPS/SSPS

We first describe a grid-based value iteration algorithm for OAMDPs/SSPs. Grid-VI uses a set of regular grid points to approximate value functions. A regular grid with the resolution $K$ is defined as:

$$P_K = \left\{ b = (\frac{1}{K})k \mid k \in I_+^{|\Theta|}, \sum_{i=1}^{|\Theta|} k(i) = K \right\}, \quad (13)$$

where $I_+^{|\Theta|}$ is the set of $|\Theta|$-vectors of non-negative integers. $P_K$ divides $\Delta^{|\Theta|}$ into a set of equal-size sub-simplices. Figure 2 shows a sample regular grid on $\Delta^3$ with $K = 2$.

As previously done in grid-based approximation algorithms for POMDPs [Lovejoy, 1991], the value at a given belief point $b \in \Delta^{|\Theta|}$ is interpolated as using the barycentric coordinates of $b$ with respect to $P_K(b)$:

$$V_K(s, b) = \sum_{b_i \in P_K(b)} \lambda_i V_K(s, b_i), \quad (14)$$

where $P_K(b)$ is the corners of the sub simplex containing $b$, $\lambda_i \geq 0$, $\sum_{i=1}^{|\Theta|} \lambda_i = 1$, and $b = \sum_{i=1}^{|\Theta|} \lambda_i b_i$. In Figure 2, the

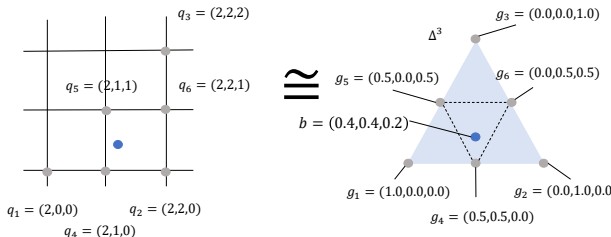

Figure 2: An example of discretized belief points $P_K$ (right) with $K = 2$ and $|\Theta| = 3$. The left is the corresponding integer points ($P'_K$).

value at $b$ is interpolated using the values at $g_4$, $g_5$, and $g_6$. For each iteration, the algorithm updates values at all $s \in S$ and $b \in P_K$ using the Bellman optimality operator ($\mathcal{T}$):

$$(\mathcal{T}V_K)(s, b) = \\ \max_{a \in A} \left[ R(s, a, b) + \gamma \sum_{s' \in S} T(s, a, s') V_K(s', b^{s,a,s'}) \right], \quad (15)$$

where values at $b \notin P_K$ are interpolated using Equation 14. The final policy is obtained by one-step lookahead using values at given belief points:

$$\pi_K(s, b) = \\ \arg\max_{a \in A} \left[ R(s, a, b) + \gamma \sum_{s' \in S} T(s, a, s') V_K(s', b^{s,a,s'}) \right]. \quad (16)$$

For problems with undiscounted objectives (OASSPs), Equation 15 is replaced with minimizing costs without the discount factor.

**Efficient Interpolation**

One key advantage of using a regular grid is that finding $\lambda$ is quite efficient. To efficiently find barycentric coordinates of $b \in \Delta^{|\Theta|}$ with respect to ($P_K(b) \subset \Delta^\Theta$), we use a Freudenthal triangulation [Freudenthal, 1942]:

$$P'_K = \left\{ q \in I_+^{|\Theta|} \mid K = q(1) \geq q(2) \geq \cdots \geq q(|\Theta|) \right\}. \quad (17)$$

Note that, we have $|P'_K| = |P_K| = \frac{(K+|\Theta|-1)!}{K!(|\Theta|-1)!}$. Due to one-to-one correspondence between points in $P_K$ and $P'_K$, we can find a barycentric coordinate for $b \in \Delta^{|\Theta|}$ using a barycentric coordinate for the corresponding $v \in I_+^{|\Theta|}$ [Lovejoy, 1991, Zhou and Hansen, 2001] as follows:

1. Given $b \in \Delta^{|\Theta|}$, let $x(i) = K \sum_{j=i}^{|\Theta|} b(\theta_j)$. For example, given $b = (0.4, 0.4, 0.2)$, we have $x = (2.0, 1.2, 0.4)$.

2. Let $v(i)$ be the largest integer such that $v(i) \leq x(i)$. In our example, $v = (2, 1, 0)$.

3. Let $d(i) = x(i) - v(i)$. In our example, $d = (0.0, 0.2, 0.4)$.

4. Let $p$ be a permutation of $1 \cdots |\Theta|$ such that $d(p(1)) \geq d(p(2)) \geq \cdots \geq d(p(|\Theta|)))$. In our example, $p = (2, 1, 0)$.

5. Identify the vertices $(v_1, v_2, \cdots, v_{|\Theta|})$ of the subsimplex in $P'_K$ containing $x$ as follows:

$$v_1 = v, \quad (18)$$

$$v_{j+1}(i) = \begin{cases} v_j(i) + 1 & \text{if } i = p(j), \\ v_j(i) & \text{otherwise.} \end{cases} \quad (19)$$

In our example, $v_1 = q_4 = (2, 1, 0)$, $v_2 = q_5 = (2, 1, 1)$, and $v_3 = q_6 = (2, 2, 1)$. Because of the one-to-one correspondence between $P_K$ and $P'_K$, this identifies the corresponding points in $P_K$ $(b_1, \cdots, b_{|\Theta|})$ containing $b$. In our example, $b_1 = g_4 = (0.5, 0.5, 0.0)$, $b_2 = g_5 = (0.5, 0.0, 0.5)$, and $b_3 = g_6 = (0.0, 0.5, 0.5)$.

6. The barycentric coordinates $\lambda_1, \cdots, \lambda_{|\Theta|}$ are determined as:

$$\lambda_i = d(p(i-1)) - d(p(i)) \text{ for } 2 \leq i \leq |\Theta|, \quad (20)$$

$$\lambda_1 = 1 - \sum_{i=2} \lambda_i. \quad (21)$$

In our example, $\lambda_1 = 0.6$, $\lambda_2 = 0.2$, and $\lambda_3 = 0.2$.

As discussed by Zhou and Hansen [2001], finding a subsimplex can be done in $\mathcal{O}(|\Theta| \log |\Theta|)$ time.

**Theoretical Guarantees**

We now discuss the theoretical guarantees of Grid-VI. Our first result shows that Grid-VI converges to the unique fixpoint in the discounted setting.

**Proposition 4.** *For an OAMDP, Grid-VI converges to the unique fixpoint $V_K^*$.*

*Proof.* The interpolation (Equation 14) can be understood as an operator on the value function. Let $\mathcal{A}_K$ be the corresponding operator, then our Grid-VI can be seen as repeatedly applying $(\mathcal{T}_K = \mathcal{A}_K \circ \mathcal{T})$ to the value function. Since $\mathcal{A}_K$ is a nonexpansion and $\mathcal{T}$ is a contraction, $\mathcal{A}_K \circ \mathcal{T}$ is also a contraction, and Grid-VI converges to the unique fixpoint $V_K^*$ [Gordon, 1995]. □

The next result establishes the error bound for the approximate value function $V_K^*$ from the optimal value function $V^*$ in the discounted setting. We first prove Lemma 1, which bounds the one-step error due to approximation.

**Lemma 1.** *For an OAMDP with Lipschitz-continuous value function with the constant $L_{V^*}$, one-step approximation errors using a regular grid with resolution $K$ are bounded as:*

$$\|\mathcal{T}_K V^* - V^*\|_\infty \leq \frac{L_{V^*}}{K}. \quad (22)$$

*Proof.* See Section A. □

In the discounted case, the overall value approximation error can be bounded using the one-step approximation error.

**Proposition 5.** *For an OAMDP whose value function is $L_{V^*}$-Lipschitz continuous, we have:*

$$\|V^* - V_K^*\|_\infty \leq \frac{L_{V^*}}{(1-\gamma)K}. \quad (23)$$

*Proof.* See Section A. □

Note that the right-hand sides go to 0 as $K \to \infty$.

Next, we show that under the standard assumptions, Grid-VI converges to the unique fixpoint when it is applied to undiscounted problems (OASSPs) as well. To prove the claim, we first note that, for an OASSP $M = \langle S, A, T, d_0, \Theta, b_0, \tau, C, G \rangle$, Grid-VI for OASSPs implicitly defines an SSP $M_K = \langle S \times P_K, A, T, d_0^K, C_K, G_K \rangle$ where

$$T_K(\langle s, b\rangle, a, \langle s', b_i\rangle) = \begin{cases} 0 & b_i \notin P_K(b^{s,a,s'}), \\ \lambda_i T(s, a, s') & b^{s,a,s'} = \sum_i \lambda_i b_i, \end{cases} \quad (24)$$

$$d_0^K(\langle s, b_i\rangle) = \begin{cases} 0 & b_i \notin P_K(b_0), \\ \lambda_i d_0(s) & b_0 = \sum_i \lambda_i b_i, \end{cases} \quad (25)$$

$$C_K(s, a, b) = C(s, a, b), \quad (26)$$

$$G_K = G \times P_K. \quad (27)$$

The states in $M_K$ consist only of the corners of subsimplices. The transitions in $M_K$ are the same as in the original OASSP, except that, after the belief update, $b^{s,a,s'}$ is transitioned to one of the belief points $b_i \in P_K(b^{s,a,s'})$ surrounding it. Note that, unlike belief MDPs corresponding to OAMDPs/SSPs, the number of belief states in $M_K$ is finite.

Since all $M$, $M_d$ and $M_K$ have the same dynamics in terms of domain state transitions, we have:

**Lemma 2.** *If $M_d$ has a proper policy, $M$ and $M_K$ also have at least one proper policy.*

*Proof.* Let $\pi_d$ be a proper policy for $M_d$. Then $\pi(\langle s, b\rangle, a) = \pi_K(\langle s, b\rangle, a) = \pi_d(s, a)$ are proper policies for $M$ and $M_K$, respectively. □

The fact that $M_K$ has a proper policy whenever $M_d$ has one lets us prove the convergence of Grid-VI.

**Proposition 6.** *If $M_d$ has a proper policy, Grid-VI for OASSPs converges to the unique fixpoint $V_K^*$.*

*Proof.* From Lemma 2, $M_K$ has a proper policy when $M_d$ has one. Our definition of OASSPs only allows positive costs. Therefore, any improper policies trivially incur infinite costs. For an SSP with a finite number of states, value iteration converges to the unique fixpoint as long as there is a proper policy and any improper policy incur infinite cost [Bertsekas and Tsitsiklis, 1991]. □

**Relationships to Grid-based Approximation Algorithms for POMDPs** Our algorithm shares similarities with grid-based approximations for POMDPs [Lovejoy, 1991, Brafman, 1997, Hauskrecht, 2000, Zhou and Hansen, 2001, Bonet, 2002]. The main difference is that the belief is over $\Theta$ in OAMDPs/SSPs instead of over $S$ as in POMDPs. Approximation using regular grids requires the number of points exponential to the dimension of belief vectors. However, in most scenarios, it is reasonable to assume that the number of possible types ($|\Theta|$) is much smaller than the number of states. Thus, having grid points exponential to the dimension of belief vectors is less of a constraint for OAMDPs/SSPs.

**Relationships to Grid-based Approximation Algorithms for Continuous MDPs** Our Grid-VI for OAMDPs/SSPs is a special case of grid-based value iteration for continuous MDPs [Chow and Tsitsiklis, 1991, Munos and Moore, 2002] in general. One main difference is that, in OAMDPs/SSPs, the continuous part of the state space ($\Delta^{|\Theta|}$) is guaranteed to be a simplex, which enables the efficient interpolation method. Another difference is that, due to the structure of OAMDPs/SSPs, discretization preserves the existence of a proper policy (Lemma 2), which helped us prove the convergence of Grid-VI for the undiscounted setting.

### 4.2 GRID-BASED REAL-TIME DYNAMIC PROGRAMMING FOR OAMDPS/SSPS

We now propose an extension of Real-Time Dynamic Programming (RTDP) [Barto et al., 1995] to OAMDPs/SSPs, called Grid-RTDP. The potential issue for Grid-VI is that it needs to update values at every state and grid points. However, many of these points could be irrelevant in computing an optimal policy. RTDP is an asynchronous value iteration algorithm that can converge to the optimal solution without having to consider the entire state space. RTDP avoids exploring a portion of the state space by utilizing an admissible heuristic, i.e. lower bounds for the expected costs to the goal. While Grid-RTDP could be applied to OAMDPs, our presentation in this section will be based on OASSPs.

Similar to Grid-VI, Grid-RTDP discretizes beliefs into regular grids. The value at a belief $b \in \Delta^{|\Theta|}$ is interpolated using Equation 14. Algorithm 1 shows a pseudocode for Grid-RTDP. The algorithm consists of repeated trials, where each trial starts from the initial state and belief of the observer. During each trial, the algorithm first maps the current belief $b$ to one of the surrounding grid points $b_i \in P_K(b)$

---

**Algorithm 1** Grid-RTDP

1: **function** GRID-RTDP
2:    **while** within computational budget **do**
3:       $TRIAL(d_0, b_0)$
4:    **end while**
5: **end function**
6:
7: **function** TRIAL($d_0, b_0$)
8:    $s \sim d_0$
9:    $b \leftarrow b_0$
10:   **while** episode continues **do**
11:      sample $b_i \in P_K(b)$ with the weight $\lambda_i$
12:      $a^* \leftarrow \min_a Q_K(s, b_i, a)$
13:      $V_K(s, b_i) \leftarrow Q_K(s, b_i, a^*)$
14:      $s' \sim \Pr(\cdot | s, a^*)$
15:      $b \leftarrow b_i^{s,a,s'}$
16:   **end while**
17: **end function**

---

randomly, where $b = \sum_{i=1}^{|\Theta|} \lambda_i b_i$. Each $b_i$ has probability $\lambda_i$ of transitioning into (line 11). Then the algorithm selects an action that minimizes the current cost estimate to the goal $Q_K(s, b_i, a)$ (line 12):

$$Q_K(s, b, a) \tag{28}$$

$$= C(s, a, b) + \sum_{s' \in S} T(s, a, s') V_K(s', b^{s,a,s'}) \tag{29}$$

$$= C(s, a, b) + \sum_{s' \in S} T(s, a, s') \sum_{b_i \in P_K(b^{s,a,s'})} \lambda_i V_K(s', b_i), \tag{30}$$

where $V_K$ is initialized with a given heuristic function $h$. In this paper, we consider the following two heuristic functions:

- $h_0$: which always returns 0 (i.e. no heuristics), and
- $h_d$: which returns the scaled optimal cost to go for the underlying domain cost ($w_d \cdot V_d^*(s)$).

Note that both $h_0$ and $h_d$ are admissible heuristics. After selecting the best action $a^*$, the cost estimate for the current state ($V_K(s, b_i)$) is updated to $Q_K(s, b_i, a^*)$ (line 34). Note that the values are updated only at beliefs in $P_k$. The next state is then sampled according to the dynamics of the environment (line 14) and the belief of the observer is updated accordingly (line 15).

**Relationships to RTDP-Bel** Grid-RTDP is akin to RTDP-Bel [Bonet and Geffner, 2009], a version of RTDP developed for POMDPs. Similar to Grid-RTDP, RTDP-Bel is based on discretizing beliefs. Let $d(b)$ be a discretization of $b$. Unlike Grid-RTDP that updates the value at $d(b)$ using Q-values at $d(b)$, RTDP-Bel updates the value at $d(b)$ using Q-values at $b$. This may cause an oscillating behavior of RTDP-Bel when two different belief points $b_1$ and $b_2$ discretize to the same point (i.e., $d(b_1) = d(b_2)$).

**Properties** We discuss some properties of Grid-RTDP. When applied to SSPs, RTDP has the following guarantee:

**Theorem 1** (Barto et al. [1995]). *If there exists a proper policy for an SSP and the initial value is admissible, then RTDP converges to the optimal value at relevant states.*

We will now show that Grid-RTDP inherits the properties analogous to Theorem 1 under the following conditions:

A1 The domain SSP $M_d$ has a proper policy.

A2 The initial value estimates are admissible.

Combining Lemma 2 with Theorem 1, we get:

**Proposition 7.** *Under A1-2, Grid-RTDP converges to the optimal values ($V_K^*$) at relevant states.*

Note that Proposition 7 proves the convergence to the optimal value for discretized problems $V_K^*$, but not to $V^*$.

### 4.2.1 Grid-Based Labeled RTDP for OASSPs

We now propose labeled RTDP (LRTDP) [Bonet and Geffner, 2002] for OASSPs, called Grid-LRTDP. The original RTDP does not explicitly check for convergence, and can keep visiting states that are already solved, resulting in its slow convergence behavior. LRTDP alleviates the issue by labeling these states as solved. The algorithm labels states as solved if residuals of Bellman updates in the states that could be visited under the current best policy are smaller than a given threshold.

Grid-LRTDP simply adds the LRTDP labeling procedure to Grid-RTDP. Alternatively, Grid-LRTDP can be understood as applying LRTDP to a discretized problem $M_K$. The pseudocode for the algorithm is available in the appendix (Appendix B).

The final policy for Grid-(L)RTDP is obtained as:

$$\pi_K(s, b, a) = \sum_{b_i \in P_K(b)} \lambda_i [a = \arg\min_{a_i \in A} Q_K(s, b_i, a_i)]. \tag{31}$$

This means we take the optimal actions at the corners of subsimplices, proportional to the corresponding weights $\lambda_i$. The reason we do not use the one-step lookahead policy (Equation 16) is that it could lead us to regions of beliefs where the values have not yet converged. When an action is selected at $b$ such that it is not optimal at any $b_i \in P_K(b)$, the updated belief might not have been checked for convergence by Grid-LRTDP.

## 5 EXPERIMENTS

We present experimental results solving OASSPs using the proposed algorithms. In the first experiment, we com-

pare Grid-VI and Grid-LRTDP on the time until values are $\epsilon$-consistent (the maximum residual is smaller than a given threshold). In the second experiment, we compare Grid-(L)RTDP and UCT in terms of their anytime performances. All the codes used in the experiments are available from `https://github.com/dosydon/approximation_algorithm_for_oamdp`.

### 5.1 DOMAINS

We briefly describe the problem domains used in the experiments.

**MazeWorld** Figure 1a shows an example of MazeWorld. The agent's goal is to reach either one of the possible goals $\{A, B, C, D, E\}$. The domain costs are proportional to the distance traveled. To encourage being clear about the intention, $C_b$ is the TV distance from the target belief. To make the problem more challenging, the agent can get transported to the initial state with probability $0.1$ at each state. $w_d = 0.1$ and $w_b = 1.0$.

**BlocksWorld** Figure 3 shows an example of Blocksworld from Miura and Zilberstein [2021], where the goal is to stack blocks to spell "ARMS". Picking up a block always succeeds with probability 1, while putting down a block fails with probability $0.3$ (the block falls on the table). Each domain action has a cost of 1. $C_b$ is the TV distance from the target belief. $w_d = 0.1$ and $w_b = 1.0$. The optimal policy first stacks "R" on top of "S". This is not optimal in terms of task progression, but tells the observer that the goal "ARMS" is more likely than "RAMS".

**Acronym** Figure 4 illustrates the Acronym domain. There are four locations with letters. The agent can move in eight different directions. Once the agent is in the locations with letters, it can toggle the letters among $A \rightarrow M \rightarrow R \rightarrow S \rightarrow A$. The potential goals are to spell "ARMS", "RAMS", or "MARS" from top left to bottom right. When toggling among letters, there is $0.3$ probability of accidentally toggling too much. The objective is to spell "ARMS" while being ambiguous about the intention. $C_b(b) = H_{max} - H(b)$ where $H_{max}$ is the entropy of the uniform distribution and $H(b)$ represents the entropy of $b$. $w_d = 0.5$ and $w_b = 1.0$.

### 5.2 COMPUTING $Q_\theta^*$

Using the Boltzmann action model (Equation 3) for the belief update (Equation 1) requires computing $Q_\theta^*$ at each state for each $\theta \in \Theta$. For Grid-VI, we used Value Iteration to compute $Q^*$ since Grid-VI needs to enumerate all states. For Grid-LRTDP and UCT, we used LRTDP from $s \in S$ as needed to compute $Q_\theta^*(s, a)$ to avoid generating the entire state space. The running times for each algorithm include the running times for computing $Q_\theta^*$.

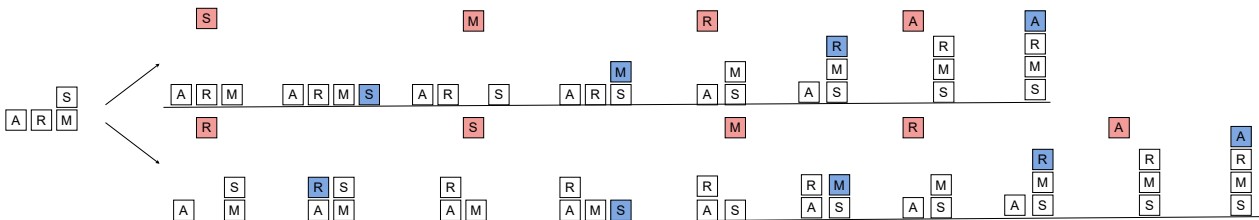

Figure 3: Task optimal behavior (top) and legible behavior (bottom) in Blocks World from [Miura and Zilberstein, 2021] Red blocks are the ones the agent is holding. Blue blocks represent blocks that were just put down. The possible goals are "ARMS" or "RAMS".

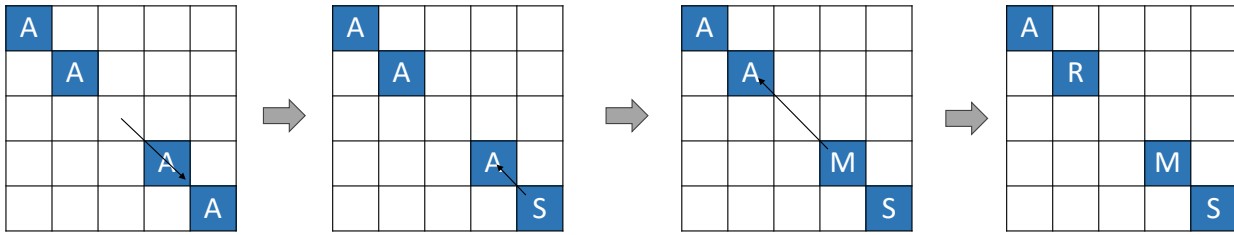

Figure 4: Acronym

## 5.3 OFFLINE CONVERGENCE

We compare the following algorithms on the time before the maximum residual is smaller than $\epsilon = 10^{-3}$:

- Grid-VI with $K = 1, 4, 8$;
- Grid-LRTDP with $K = 1, 4, 8$ using $h_0$ and $h_d$.

Each run has time limit 10m and memory limit 8Gbytes.

Table 1 shows the results. Grid-LRTDP using $h_d$ was overall the best algorithm, generating fewer belief states to solve problems. The exception was the MazeWorld domain, where, due to the random transition back to the initial state, Grid-(L)RTDP had to generate most of the belief points. While some problems required only coarse discretization of beliefs, other problems required finer discretization to compute near optimal policies.

## 5.4 ANYTIME PERFORMANCE

We compare the following algorithms in terms of the anytime behaviors:

- Grid-(L)RTDP with $K = 4, 8$ using $h_d$;
- UCT where the rollout policy $\pi_d^*$ is an optimal policy for the domain SSP.

Each algorithm was run for $10^2, 10^3, 5 \cdot 10^3, 10^4, 5 \cdot 10^4,$ $10^5, 5 \cdot 10^5, 10^6$ Grid-(L)RTDP/UCT trials. For UCT, the

specified number of trials are performed at each timestep online. For Grid-(L)RTDP, the trials are performed offline. Since both UCT and Grid-LRTDP before convergence do not necessarily reach the goal state, both of the algorithms are evaluated on the average costs for the first 50 time steps. Each run has time limit 10m and memory limit 2Gbytes.

Figure 5 shows the results. UCT and Grid-(L)RTDP exhibited performances that complement each other. While UCT showed better anytime performance in Acronym, it took some time to achieve good performance in Blocks World, a small problem instance with $|\Theta| = 2$. Comparing Grid-(L)RTDP with different resolutions ($K$), using coarser grids generally resulted in better anytime behaviors as long as the resolution is sufficient. Between Grid-RTDP and Grid-LRTDP, they exhibited comparable anytime behaviors.

## 6 RELATED WORK

OAMDP is a framework that unifies different kinds of observer-aware behaviors. Observer-aware behaviors include *Legible* behavior [Dragan and Srinivasa, 2013, Miura et al., 2021], which implicitly conveys intentions via the choice of actions. Similarly, *explicable* behaviors [Zhang et al., 2017, Gong and Zhang, 2022] conform to observers' expectations. *Deceptive* behaviors [Dragan et al., 2015, Masters and Sardina, 2017, Savas et al., 2022] hide agents' intentions or actively deceive observers. *Predictable* behaviors enable observers to predict future actions [Fisac et al.,

| Domain | $|\Theta|$ | K | Grid-VI | | | | Grid-LRTDP($h_0$) | | | | Grid-LRTDP($h_d$) | | | |
|---|---|---|---|---|---|---|---|---|---|---|---|---|---|---|
| | | | V | t(s) | $|S|$ | $|P_K|$ | V | t(s) | $|S|$ | $|P_K|$ | V | t(s) | $|S|$ | $|P_K|$ |
| MazeWorld | 5 | 1 | 18.90 | 18.07 | 148 | 740 | 19.14 | 3.79 | 148 | 739 | 19.11 | 3.95 | 148 | 602 |
| | | 4 | - | - | - | - | 17.08 | 208.35 | 148 | 10164 | 16.98 | 259.28 | 148 | 9711 |
| | | 8 | - | - | - | - | - | - | - | - | - | - | - | - |
| Acronym | 3 | 1 | 15.25 | 15.25 | 6379 | 19137 | 15.76 | 9.93 | 6379 | 19137 | 15.72 | 9.95 | 6379 | 19137 |
| | | 4 | 7.89 | 346.90 | 6379 | 95685 | 8.63 | 38.91 | 6379 | 88765 | 8.70 | 31.81 | 6379 | 81728 |
| | | 8 | - | - | - | - | 8.43 | 221.75 | 6379 | 286953 | 8.48 | 165.40 | 6379 | 265752 |
| BlocksWorld | 2 | 1 | 3.67 | 3.98 | 125 | 250 | 3.56 | 3.34 | 125 | 250 | 3.58 | 1.72 | 125 | 134 |
| | | 4 | 3.13 | 10.02 | 125 | 625 | 3.04 | 6.99 | 125 | 543 | 3.03 | 5.24 | 124 | 392 |
| | | 8 | 3.14 | 17.36 | 125 | 1125 | 3.04 | 12.0 | 125 | 890 | 3.04 | 8.85 | 123 | 625 |

Table 1: Time until convergence for different algorithms. $V$ represents the value when the policy is evaluated under the true environment ($M$). $t(s)$ is the running time in seconds. $|S|$ and $|P_K|$ represent the number of generated domain and belief states, respectively.

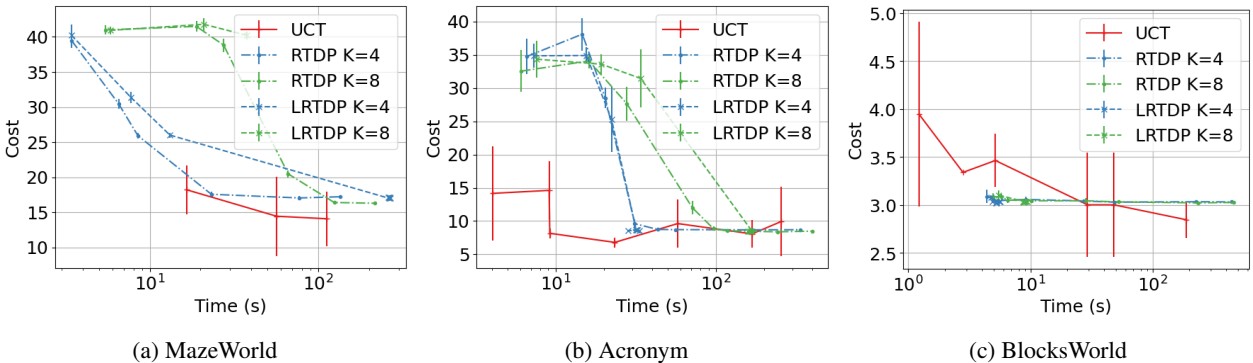

(a) MazeWorld       (b) Acronym       (c) BlocksWorld

Figure 5: Anytime behaviors for different algorithms.

2020, Lepers et al., 2024]. Agents can also express their *(in)capability* via the choice of their actions [Kwon et al., 2018]. OAMDP can also model the combination of implicit communication through behaviors and explicit communication with messages [Miura and Zilberstein, 2024].

OAMDP could be regarded as a special case of Decision Process with non-Markovian Reward (NMRDP) [Bacchus et al., 1996, Thiébaux et al., 2006]. Unlike OAMDPs, existing works on NMRDPs Bacchus et al. [1996], Thiébaux et al. [2006], Brafman et al. [2018] utilize temporal logic to describe rewards over histories. OAMDP, on the other hand, employs the belief of the observer to capture the non-Markovian nature of rewards.

Recent years have seen a surge of interest in the human tendency to ascribe intentionality to autonomous agents [Thellman et al., 2017, Perez-Osorio and Wykowska, 2020]. In other words, humans often interpret the behaviors of autonomous agents as rational behaviors driven by intentions, beliefs, and desires. While people do not necessarily understand the internal mechanisms of the agents, people can still predict the behaviors of the agents by ascribing intentionality to them. OAMDPs rely on the tendency to take intentional stances to autonomous agents.

## 7 CONCLUSION

In this paper, we propose the first approximation algorithms for solving OAMDPs/SSPs, Grid-VI and Grid-(L)RTDP. Both of the algorithms are based on discretizing the observer's beliefs into regular grids. To justify the proposed algorithms, we show that the domain state and the belief of the observer constitute a sufficient statistics for OAMDPs (Proposition 1). Furthermore, we show that both algorithms converge to the unique value (Proposition 4, 6, and 7) and provide performance guarantees under the standard assumptions (Propositions 5 and 7). Our experimental results show that the proposed algorithms can compute near-optimal policies for OAMDPs/SSPs. In particular, Grid-(L)RTDP can converge to a solution faster than Grid-VI and has anytime performance competitive with UCT.

## 8 ACKNOWLEDGEMENTS

This research was supported in part by the NSF grant IIS-2205153 and by the Alliance Innovation Lab Silicon Valley.

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

# A  PROOFS

To prove Proposition 2, we first prove the Lipschitz continuity of $n$-step value function. Let $V^{(0)}(s,b) = 0$ and $V^{(n+1)}(s,b) = \max_a R(s,a,b) + \gamma \sum_{s'} T(s,a,s')V^{(n)}(s',b^{s,a,s'})$. Then we have:

**Lemma 3.** *For a $(L_r, L_p)$-Lipschitz OAMDP, $V^{(n)}$ is $L_{V^{(n)}}$-Lipschitz continuous, where $L_{V^{(n)}}$ satisfies:*

$$L_{V^{(n+1)}} = L_r + \gamma L_p L_{V^{(n)}} \tag{32}$$

*Proof.* Proof by induction on $n$. For the base case with $n = 1$,

$$|V^{(1)}(s,b_1) - V^{(1)}(s,b_2)| \tag{33}$$
$$= |\max_a R(s,a,b_1) - \max_a R(s,a,b_2)| \tag{34}$$
$$\leq \max_a |R(s,a,b_1) - R(s,a,b_2)| \tag{35}$$
$$\leq L_r \|b_1 - b_2\|_\infty \tag{36}$$

For the induction step,

$$|V^{(n+1)}(s, b_1) - V^{(n+1)}(s, b_2)| \qquad (37)$$

$$= |\max_a R(s, a, b_1) + \gamma \sum_{s'} T(s, a, s') V^{(n)}(s', b_1^{s,a,s'}) \qquad (38)$$

$$- \max_a R(s, a, b_2) + \gamma \sum_{s'} T(s, a, s') V^{(n)}(s', b_2^{s,a,s'})| \qquad (39)$$

$$\leq \max_a |R(s, a, b_1) + \gamma \sum_{s'} T(s, a, s') V^{(n)}(s', b_1^{s,a,s'}) \qquad (40)$$

$$- R(s, a, b_2) + \gamma \sum_{s'} T(s, a, s') V^{(n)}(s', b_2^{s,a,s'})| \qquad (41)$$

$$\leq \max_a |R(s, a, b_1) - R(s, a, b_2)| \qquad (42)$$

$$+ \gamma \sum_{s'} T(s, a, s') |V^{(n)}(s', b_1^{s,a,s'}) - V^{(n)}(s', b_2^{s,a,s'})| \qquad (43)$$

$$\leq (L_r + \gamma L_p L_{V^{(n)}}) \|b_1 - b_2\|_\infty \qquad (44)$$

$$\square$$

**Proposition 2.** *For a $(L_r, L_p)$-Lipschitz OAMDP, if $\gamma L_p < 1$, then $V^*$ is $L_{V^*}$-Lipschitz continuous where:*

$$L_{V^*} = \frac{L_r}{1 - \gamma L_p}. \qquad (12)$$

*Proof.* Consider a sequence $\{L_n\}_{n \geq 1}$ where $L_1 = L_r$ and:

$$L_{n+1} = L_r + \gamma L_p L_n \qquad (45)$$

Then,

$$L_n = L_r + \gamma L_p L_r + (\gamma L_p)^2 L_r + \cdots + (\gamma L_p)^{n-1} L_r \qquad (46)$$

$$= \frac{1 - (\gamma L_p)^n}{1 - \gamma L_p} L_r \qquad (47)$$

By our assumption, $\gamma L_p < 1$, so the sequence converges. Let $L_{V^*} = \lim_{n \to \infty} L_n$. $L_{V^*}$ must satisfy $L_{V^*} = L_r + \gamma L_p L_{V^*}$. Thus, we get Equation 12. $\square$

**Proposition 3.** *If $\tau^{s,a,s'}(\theta) > 0$ for all $\theta \in \Theta$, $s, s' \in S$, and $a \in A$, belief transitions are Lipschitz continuous.*

*Proof.* Let $f^{s,a,s'}(b) = b^{s,a,s'} : \Delta^\Theta \to \Delta^\Theta$ be the belief transition after observing $\langle s, a, s' \rangle$. From the definition (Equation 1), $f^{s,a,s'}(b)(\theta_i) = \frac{\tau_i^{s,a,s'} b_i}{\sum_k \tau_k^{s,a,s'} b_k}$, where

$\tau_i^{s,a,s'} = \tau^{s,a,s'}(\theta_i)$ and $b_i = b(\theta_i)$. Then we have:

$$J_{f^{s,a,s'}}(b)_{i,j} = \begin{cases} \frac{\tau_i^{s,a,s'}(\sum_{k \neq i} \tau_k^{s,a,s'} b_k)}{(\sum_k \tau_k^{s,a,s'} b_k)^2} & i = j, \\ \frac{-\tau_i^{s,a,s'} \tau_j^{s,a,s'} b_j}{(\sum_k \tau_k^{s,a,s'} b_k)^2} & i \neq j, \end{cases} \qquad (48)$$

$$\|J_{f^{s,a,s'}}(b)\|_\infty = \max_{1 \leq i \leq n} \sum_{1 \leq j \leq n} |J_{f^{s,a,s'}}(b)_{i,j}|, \qquad (49)$$

$$= \max_{1 \leq i \leq n} \frac{2\tau_i^{s,a,s'}(\sum_{k \neq j} \tau_k^{s,a,s'} b_k)}{(\sum_k \tau_k^{s,a,s'} b_k)^2}, \qquad (50)$$

where $J_f$ is the Jacobian of $f$ and $\|\cdot\|_\infty$ is the induced operator norm. Let $\tau_{\min} = \min_{s,a,s',k} \tau_k^{s,a,s'}$ and $\tau_{\max} = \max_{s,a,s',k} \tau_k^{s,a,s'}$. Note that, for every $b \in \Delta^n$, $\sum_{k \neq i} \tau_k^{s,a,s'} b_k \leq \tau_{\max}$ and $\sum_k \tau_k^{s,a,s'} b_k \geq \tau_{\min} > 0$. Then we get $\|J_{f^{s,a,s'}}(b)\|_\infty \leq 2(\frac{\tau_{\max}}{\tau_{\min}})^2$. $\square$

**Lemma 1.** *For an OAMDP with Lipschitz-continuous value function with the constant $L_{V^*}$, one-step approximation errors using a regular grid with resolution $K$ are bounded as:*

$$\|\mathcal{T}_K V^* - V^*\|_\infty \leq \frac{L_{V^*}}{K}. \qquad (22)$$

*Proof.* For all $n \geq 0$, $K \geq 1$, $s \in S$, and $b \in \Delta^{|\Theta|}$,

$$|V^*(s, b) - \mathcal{T}_K V^*(s, b)| \qquad (51)$$

$$= |V^*(s, b) - \sum_{b_i \in P_K(b)} \lambda_i \mathcal{T} V^*(s, b_i)| \text{ (by definition)} \qquad (52)$$

$$= |\sum_{b_i \in P_K(b)} \lambda_i (V^*(s, b) - V^*(s, b_i))| \text{ ($\mathcal{T}$ is a fixpoint of $V^*$)} \qquad (53)$$

$$\leq \sum_{b_i \in P_K(b)} \lambda_i |V^*(s, b) - V^*(s, b_i)| \text{ (triangle inequality)} \qquad (54)$$

$$\leq \sum_{b_i \in P_K(b)} \lambda_i L_{V^*} \|b - b_i\|_\infty \qquad (55)$$

$$\leq L_{V^*} \frac{1}{K} \qquad (56)$$

$$\square$$

**Proposition 5.** *For an OAMDP whose value function is $L_{V^*}$-Lipschitz continuous, we have:*

$$\|V^* - V_K^*\|_\infty \leq \frac{L_{V^*}}{(1 - \gamma)K}. \qquad (23)$$

*Proof.*

$$\|V^* - V_K^*\|_\infty \tag{57}$$

$$\leq \|V^* - \mathcal{T}_K V^* + \mathcal{T}_K V^* - V_K^*\|_\infty \tag{58}$$

$$\leq \|V^* - \mathcal{T}_K V^*\|_\infty + \|\mathcal{T}_K V^* - \mathcal{T}_K V_K^*\|_\infty \tag{59}$$

$$\leq \frac{L_{V^*}}{K} + \gamma\|V^* - V_K^*\|_\infty \tag{60}$$

$\square$

# B PSEUDOCODE FOR GRID-LRTDP

Algorithm 2 shows the pseudocode for Grid-LRTDP. The algorithm operates identically to Grid-RTDP, except that at the end of each trial, the algorithm checks if states visited during the trial can be labeled as solved.

---
**Algorithm 2** Grid-LRTDP
---
1: **function** GRID-LRTDP($s_0, b_0, \epsilon, K$)
2:  **while** $\exists b_i \in P_K(b_0) \neg\langle s_0, b_0\rangle.solved$ **do**
3:   LRTDPTRIAL($s_0, b_0, \epsilon, K$)
4:  **end while**
5: **end function**
6:
7: **function** LRTDPTRIAL($s_0, b_0$)
8:  $visited \leftarrow Stack :: new()$
9:  $s \sim s_0$
10:  $b \leftarrow b_0$
11:  **while** episode continues **do**
12:   sample $b_i \in P_K(b)$ with the weight $\lambda_i$
13:   $visited.push(\langle s, b_i\rangle)$
14:   $a^* \leftarrow \min_a Q_K(s, b_i, a)$
15:   $V_K(s, b_i) \leftarrow Q_K(s, b_i, a^*)$
16:   $s' \sim \Pr(\cdot|s, a^*)$
17:   $b \leftarrow b_i^{s,a,s'}$
18:  **end while**
19:
20:  **while** $\neg visited.is\_empty()$ **do**
21:   $\langle s, b\rangle \leftarrow visited.pop()$
22:   **if** $\neg$ CHECKSOLVED($s, b, \epsilon, K$) **then**
23:    **break**
24:   **end if**
25:  **end while**
26: **end function**

---

Algorithm 3 shows the procedure for labeling states. Starting from a given $\langle s, b\rangle$ the algorithm visits state that could be visited under the current best policy, and checks if the residuals of Bellman updates are smaller than a given threshold $\epsilon$.

---
**Algorithm 3** CHECKSOLVED
---
1: **function** CHECKSOLVED($s, b, \epsilon, K$)
2:  $rv \leftarrow true$
3:  $open \leftarrow Stack :: new()$
4:  $closed \leftarrow Stack :: new()$
5:  **if** $\neg\langle s, b\rangle.solved$ **then**
6:   $open.push(\langle s, b\rangle)$
7:  **end if**
8:  **while** $\neg open.is\_empty()$ **do**
9:   $\langle s, b\rangle \leftarrow open.pop()$
10:   $closed.push(\langle s, b\rangle)$
11:   $a^* \leftarrow \min_a Q_K(s, b, a)$
12:   $\epsilon_{res} \leftarrow |V_K(s, b) - Q_K(s, b, a^*)|$
13:   $V_K(s, b) \leftarrow Q_K(s, b, a^*)$
14:   **if** $\epsilon_{res} < \epsilon$ **then**
15:    **continue**
16:   **end if**
17:   **for all** $s' \in S$ such that $T(s, a, s') > 0$ **do**
18:    **for all** $b_i \in P_K(b^{s,a,s'})$ such that $\lambda_i > 0$ **do**
19:     **if** $\neg\langle s', b_i\rangle.solved \wedge \neg\langle s', b_i\rangle \in open \wedge \neg\langle s', b_i\rangle \in closed$ **then**
20:      $open.push(\langle s', b_i\rangle)$
21:     **end if**
22:    **end for**
23:   **end for**
24:  **end while**
25:
26:  **if** $rv = true$ **then**
27:   **for all** $\langle s, b\rangle \in closed$ **do**
28:    $\langle s, b\rangle.solved \leftarrow true$
29:   **end for**
30:  **else**
31:   **while** $\neg closd.is\_empty()$ **do**
32:    $\langle s, b\rangle \leftarrow open.pop()$
33:    $a^* \leftarrow \min_a Q_K(s, b, a)$
34:    $V_K(s, b) \leftarrow Q_K(s, b, a^*)$
35:   **end while**
36:  **end if**
37: **end function**

---