# OpenReview forum: "Approximation Algorithms for Observer Aware MDPs"
_auai.org/UAI/2024/Conference — UAI 2024 poster_

### Official Review · Reviewer_EWKV · 2024-03-10

**Q2-1 Originality-Novelty:** 3
**Q2-2 Correctness-Technical Quality:** 2
**Q2-5 Clarity Of Writing:** 4

**Q10 Ethical Concerns:**

No, the contributions are proofs and new algorithms for existing problems.

**Q1 Summary And Contributions:**

The paper considers observer aware MDPs: these MDPs include a belief function that depends on the actions taken by the agent, and the reward function depends on both the current state and the belief. As such, these MDPs can be used to find strategies where an agent tries to be clear about its goal, or on the contrary tries to hide it. Two different objectives are studied in the paper: discounted sum, and stochastic shortest path.

The paper presents two algorithms to find strategies in OAMDPs: both rely on a discretization of the belief space and an interpolation of the belief. The first algorithm follows value-iteration, relying on the assumption that the value function is Lipschitz-continuous. The second
algorithm relies on Real-Time dynamic programming, that aims at only visiting relevant parts of the belief space. Both techniques are implemented, and compared with an existing algorithm.

While discretization and RTDP are not new by themselves, the paper shows in detail that the setting of OAMDPs has several specificities that
makes them very efficient: the continuous space is a simplex, and is often low dimension compared to the state space. They also propose
changes to RTDP, like checking convergence and avoiding oscillating behavior.

I read the proofs in detail and did not find any flaw; they make very good use of the assumption of Lipschitz continuity.

The definitions took me some time to follow at first, probably because of space constraints, but I could read the paper without any
problem after this. I think that there are a few minor definition problems that I detail below.

I would have liked a reproducibility for the code, to be able to check the values given. Since the experimental results are not the only focus
of the paper, this is not critical

**Q2-3 Extent To Which Claims Are Supported By Evidence:**

4: Excellent: all claims are supported by very convincing evidence (in the form of comprehensive experimental evaluation, rigorous mathematical proofs, detailed (pseudo-)code, precise references, well-motivated and realistic assumptions) and the authors deliver what they promise.

**Q2-4 Reproducibility:**

3: Good: key resources (e.g. proofs, code, data) are available and key details (e.g. proofs, experimental setup) are sufficiently well-described for competent researchers to confidently reproduce the main results.

**Q3 Main Strengths:**

- Lipschitz-continuous rewards/beliefs are used very well
- The two algorithms proposed have interesting tradeoff
- Experimental results and comparison with meaningful existing algorithms

**Q4 Main Weakness:**

- Lack of reproducibility package for the code

**Q5 Detailed Comments To The Authors:**

I have a few details that I think need to be clarified in the
definitions, since they seem either to have minor errors, or the
explanation is not detailed enough:

- Is having a proper policy sufficient? Bertsekas and Tsitsiklis, 1991
seem to also assume that "each improper stationary policy yields
infinite cost for at least one initial state", including for the result
you use to get Proposition 6. Barto et al. 1995 also seem to use this
assumption, and you use their result for Theorem 1.

- What do you mean by " the probability of St+1=st+1 when At=at and
St=st"? Do you mean "the probability of st+1\in S when at \in A and
st \in S"? Since St+1, At, and St+1 haven't been introduced, it was
confusing.

- "s0 ∼ d0" I can guess that s0 is meant to be the initial state, since
it is sampled from d0, but it should be clearly stated.

- S,A have to be finite: can you always define the expectation of the
reward or cost if S and A are both infinite? I think there are edge
cases where it is not the case, and so you may want to assume finiteness
of S and A from the start.

- b0 ∈ ∆^|Θ| is a simplex on Θ: I think it should be a simplex over
R^|Θ| or Q^|Θ|? Since Θ doesn't necessarily have the geometrical
structure to have a simplex, as defined. Also, as far as I understand,
it should not be any simplex, but the specific simplex defined by (0)
and every unit in each dimension. You seem to already mainly use it like
this.

**Q9 Complying With Reviewing Instructions:**

Yes

---

> ### Author Rebuttal · Authors · 2024-04-04
>
> Thank you for your thoughtful and thorough review of our work. We are pleased that you find the theoretical contributions solid, the algorithms well-justified for OAMDPs, and the experimental results insightful. We appreciate you taking the time to carefully check the proofs. Below we address the main points you raised.
>
> Regarding the definitions:
>
> For the conditions of Bertsekas and Tsitsiklis (1991), our definition of (OA)SSPs only allows strictly positive costs per timestep. Thus, the conditions “improper stationary policy yield infinite cost for at least one initial state” are trivially satisfied. However, this point should have been explicitly stated in the paper. We will clarify this point in the revision.
>
> As for the notations for the transition probability, S_t and A_t are random variables for states and actions at time step t, respectively. We will clarify the notation for the transition probability P(s' | s, a) and state the initial state distribution d0 unambiguously.
>
> As for the finiteness of S and A, we will make this assumption clearer in the revised version.
>
> Thanks for pointing out the notational issues with beliefs over types. We will clarify that b0 is an element of the probability simplex over R^|Θ|.
>
> Reproducibility package: We agree that having code available is valuable for reproducibility. We will prepare a code package with instructions to reproduce the experiments and make it available online upon publication.
>
> Your positive assessment of the overall novelty, correctness, and clarity is greatly appreciated. We are confident we can address the minor definition issues you raised without impacting the core results.

---

### Official Review · Reviewer_Eeqw · 2024-03-20

**Q2-1 Originality-Novelty:** 2
**Q2-2 Correctness-Technical Quality:** 3
**Q2-5 Clarity Of Writing:** 3

**Q10 Ethical Concerns:**

No ethical concerns

**Q1 Summary And Contributions:**

The paper presents approximation algorithms for Observer Aware Markov Decision Processes (OAMDPs). This model is used for situations in which the rewards depend on the beliefs of an observer about the agent's type (which is fixed but unknown), in addition to state-based rewards. The authors propose two grid based algorithms, Grid-VI (Value iteration) and Grid-RTDP (Real-Time Dynamic Programming) for OAMDPs and OASSPs (SSP variant of OAMDPs). These algorithms come with theoretical guarantees on convergence and optimality gaps (for OAMDPs).

**Q2-3 Extent To Which Claims Are Supported By Evidence:**

3: Good: the main claims are supported by convincing evidence (in the form of adequate experimental evaluation, proofs, (pseudo-)code, references, assumptions).

**Q2-4 Reproducibility:**

2: Fair: key resources (e.g. proofs, code, data) are unavailable but key details (e.g. proof sketches, experimental setup) are sufficiently well-described for an expert to confidently reproduce the main results.

**Q3 Main Strengths:**

- The paper is mostly well written. The algorithms, lemmas and proofs are presently clearly.
- The algorithms come with theoretical guarantees on convergence and optimality gaps. This just requires Lipschitz conditions on the OAMDP model. Further, these Lipschitz conditions are shown to be satisfied by a large number of models (which just require the observer to be using a noisy rational model).
- The Grid-RTDP algorithm is a valuable extension as it helps speed up the planning.

**Q4 Main Weakness:**

- The paper does not motivate the need for OAMDPs very well. The model seems very close to that of a POMDP where the part of the state space is completely observable and the rest hidden.

- The algorithms and results seem to be an application of existing work for continuous MDPs or POMDP to the setting of OAMDPs. The paper mentions differences with this prior work but not discuss the technical challenges faced or new contributions made.

- It is not clearly/explicitly mentioned that the algorithms for OASSPs have only convergence guarantees but no optimality guarantees. Also, the Grid-RTDP seems to have theoretical guarantees for the SSP setting. It is not clear if the same would extend to the discounted setting.

- The writing in the experimental section has to be improved. The descriptions of the experiments aren't clear. Also, it is difficult to read and understand Figure 4.

**Q5 Detailed Comments To The Authors:**

Some minor comments and suggestions are:

- Please provide a more sound definition of a policy.
- Describe a few real life scenarios which cannot be well accommodated by a POMDP model but OAMDP.
- Define what is meant by admissible heuristics and include the heuristic choice step in Algorithm 1.
- Any reason not to include experiments for the OAMDP discounted setting ?

**Q9 Complying With Reviewing Instructions:**

Yes

---

> ### Author Rebuttal · Authors · 2024-04-04
>
> Thank you for your detailed review and assessment of our work. We appreciate your positive comments on the theoretical guarantees, clarity of the algorithms and proofs, and the value of Grid-RTDP. Below we address some of the main weaknesses you identified.
>
> Motivating OAMDPs and the difference from POMDPs: We agree that the motivation behind OAMDPs and the difference from POMDPs could be better articulated in the paper.  While OAMDPs share similarities with POMDPs, there is a fundamental difference between the two modes: In OAMDPs, the belief is the observer's belief over the agent's possible types, while in POMDPs, it's the acting agent's belief over possible world states. This leads to distinct belief update dynamics. In OAMDPs, observations don't provide additional information to the acting agent, but actions serve as evidence of the agent's type to the observer. One may try formulating the problem as a POMDP, where the state space is a pair of physical states and types the observer believes in ($S \times \Theta$). However, if we make the belief update of the observer as a part of the transition function, we end up with the myopic observer, who changes their belief based only on the current information. Consider, for example, an autonomous vehicle (AV) trying to inform its passengers about its capabilities. In a POMDP formulation, whether or not the passenger thinks the AV is capable is a binary variable in the state space, and it stochastically transitions based only on the information in the current state (they could completely forget a mistake the AV made in the past). In OAMDPs, the belief of the passenger is in the state space, which is a summary of their observation so far. Please refer to previous work on OAMDPs for more detailed discussions on the difference from POMDPs (https://proceedings.mlr.press/v161/miura21a.html). We will clarify the difference between OAMDPs and POMDPs and cite the relevant work. Moreover, we will better describe the motivations behind OAMDPs in the introduction.
>
> Novelty over prior work: Since OAMDPs and POMDPs formulate different problems, the two problems have different structures. In particular, OAMDPs do not have piece-wise linear convexity (PWLC) of value functions, which most POMDP algorithms are based on. Therefore, algorithms for POMDPs based on PWLC do not apply to OAMDPs. As suggested, we will make this technical challenge more explicit in the revised version.
> Unlike POMDPs or continuous MDP, however, OAMDPs have the unique property that the physical state transitions and termination conditions are solely determined by the discrete part of the state space. This property ensures the existence of a proper policy for discretized problems (Lemma 2) and allows us to guarantee convergence (Proposition 6). It is important to note that this property does not necessarily hold for discretized POMDPs and continuous MDPs. In these problems, the transition dynamics are dependent on the discretized states, which may not perfectly capture the original transition dynamics. In the revised manuscript, we will emphasize this distinction and explain how our algorithmic innovations are specifically tailored to exploit the unique characteristics of OAMDP/SSPs.
>
> Clarifying theoretical results: Thank you for catching the lack of clarity on the OASSP results. You are correct that for OASSPs with undiscounted objectives, we have convergence but not approximation bounds, while for discounted OAMDPs we have both. As for the theoretical guarantees of Grid-RTDP for the discounted setting, it is known every discounted MDP can be translated into an equivalent undiscounted problem. Thus, we can have the same guarantee for the discounted problems by first translating them to equivalent undiscounted problems. We will state these facts explicitly in the revision.
>
> Improving experimental description: We will revise this section to better explain the experimental setup and results. Figure 4 will be updated for clarity.
>
> Other detailed comments:
> We will provide a more rigorous definition of a policy as a mapping from states to distributions over actions.
> We added the AV example to highlight the difference from POMDPs.
> Admissible heuristics will be defined and the heuristic will be added as an input to Algorithm 1.
> Due to space limitations, we did not have enough space to report experimental results for both discounted and undiscounted settings. OASSP was chosen since a cost-based formulation is more common in the context of heuristic search.
>
> We believe we can address your main concerns regarding motivation, novelty claims, and theoretical/experimental clarity in the revision without impacting the core technical content. The other points relating to writing and organization will also be incorporated.

---

### Official Review · Reviewer_6cou · 2024-03-22

**Q2-1 Originality-Novelty:** 3
**Q2-2 Correctness-Technical Quality:** 3
**Q2-5 Clarity Of Writing:** 3

**Q1 Summary And Contributions:**

In this manuscript, approximation algorithms are proposed for Markov decision processes (MDPs) and observer awareness (OAMDPs). In these, rewards can depend on assumed beliefs, based on observations. The algorithms apply also to stochastic shortest path problems (SSPs). Two approximation algorithms are proposed, Grid-VI and Grid-RTPD.  In the paper, theoretical properties are presented for the algorithms, like convergence guarantess and bounds on errors, and an experimental evaluation is carried out, showing promise of the proposed algorithms.

**Q2-3 Extent To Which Claims Are Supported By Evidence:**

2: Fair: the main claims are somewhat supported by evidence (but the experimental evaluation may be weak, or does not match entirely with the claims, important baselines may be missing, proofs contain important ideas but lack rigor, algorithmic details are only discussed superficially, references are imprecise, assumptions are not sufficiently motivated or explicated, etc.).

**Q2-4 Reproducibility:**

2: Fair: key resources (e.g. proofs, code, data) are unavailable but key details (e.g. proof sketches, experimental setup) are sufficiently well-described for an expert to confidently reproduce the main results.

**Q3 Main Strengths:**

I would say that a strength of the manuscript is that both theoretical foundations are studied for the proposed algorithms and an experimental evaluation is performed to show impact of the approximation algorithms. The problems studied, that of OAMDPs and SSPs, appear sufficiently relevant to me.

I think that the presentation is adequate for the presented material and content. Due to space limitations, I do not see much potential for improvement, even though I would have appreciate a bit more examples.

I am not an expert in the area of the manuscript, but as a somewhat outside view, I think that the presented contributions are sufficiently significant for a positive recommendation of the manuscript: the paper proposed to novel approximation algorithms, provides clear theoretical guarantess and properties, discusses limitations, and the empirical evaluation shows strengths of the proposed algorithms (however note a question/weakness below).

**Q4 Main Weakness:**

From a technical perspective, I think that some concepts used in the manuscript and proofs steps require a bit more background than provided. I do not see this as a critical problem; the manuscript refers in several places to related results to establish the current results. Again, due to space limitations and I do not have concrete suggestions for improvement here.

As a question to the author(s), I did not see a clear discussion or references whether the used instances in the experiments are part of a standard benchmark set or are generated by the author(s). Did you use known instances? If not, how did you generate them? The running times are reasonable, but not very high, this might be a potential weakness in that the instances are not of sufficient difficulty. I would ask the author(s) to respond. Also, how many instances were used per domain? I also think that no machine details are provided.

**Q5 Detailed Comments To The Authors:**

A question is phrased in the "Weaknesses" section above.

Minor comments
-abstract: Grid-Vi => Grid-VI
-Introduction: "Proposition5"
-Section 3.4: different symbol for expectation
-Figure 4: not a problem for me, but in black/white this figure is difficult to interpret (possibly different uses of dashes/dotted/... may help).

**Q9 Complying With Reviewing Instructions:**

Yes

---

> ### Author Rebuttal · Authors · 2024-04-04
>
> Thank you for your thoughtful review of our manuscript. We are pleased that you find our algorithms, theoretical analysis, and empirical evaluation to make a sufficiently significant contribution. Below we address some of your main comments and questions.
>
> Regarding the technical background: We agree that due to the page limit, some technical details and background had to be condensed. We will revise the paper to include additional intuition and refer more explicitly to the key-related works for readers less familiar with the area.
>
> Experimental setup details: Thank you for bringing this to our attention. One of the problems we used, BlocksWorld, was sourced from the work of Miura et al. (2021). However, due to the lack of a standard benchmark for OAMDPs and OASSPs, we manually designed the other instances used in our experiments. We acknowledge that the instances are relatively small to moderate in size, as our primary objective was to initially assess the performance of the algorithms on problems that were reasonably manageable. While we reported the results for a single representative instance per domain, we observed comparable patterns across other instances as well. To provide more clarity, we will incorporate these specifics into the description of our experimental setup.
>
> Machine specs: The experiments were performed on a machine with Intel(R) Xeon(R) Gold 6126 CPU.
>
> Minor comments: We appreciate you pointing out these small errors and areas for improvement. We will fix the typos and look into improving Figure 4's legibility in grayscale.
>
> Your perspective as a reviewer somewhat outside the area is highly valuable. We believe the concerns you raise can be readily addressed in a revision without impacting the core contributions.

---

### Official Review · Reviewer_6BWL · 2024-03-23

**Q2-1 Originality-Novelty:** 3
**Q2-2 Correctness-Technical Quality:** 3
**Q2-5 Clarity Of Writing:** 4

**Q10 Ethical Concerns:**

No ethical concerns are raised.

**Q1 Summary And Contributions:**

The paper develops approximation algorithms for solving OAMDPs (observer-aware Markov decision problems, maximization of utils) or OASSP (observer-aware stochastic shortest path problems with minimisation of costs). The two algorithms proposed, Grid-VI and Grid-(L)RTDP, are based on discretizing the observer’s beliefs (seen as points stemming from a distribution of oberserver types) into regular grids. It is shown that the domain state and the belief of the observer constitute a sufficient statistics for OAMDPs, i.e., there is no history required. Furthermore, convergence is shown. In addition, performance guarantees under standard assumptions are illustrated. Experimental results show that the proposed algorithms can compute near-optimal policies for OAMDP/SSPs. Grid-(L)RTDP can
converge to a solution faster than Grid-VI and has anytime performance competitive with the UCT algorithm.

**Q2-3 Extent To Which Claims Are Supported By Evidence:**

3: Good: the main claims are supported by convincing evidence (in the form of adequate experimental evaluation, proofs, (pseudo-)code, references, assumptions).

**Q2-4 Reproducibility:**

4: Excellent: key resources (e.g. proofs, code, data) are available and key details (e.g. proof sketches, experimental setup) are comprehensively described for competent researchers to confidently and easily reproduce the main results.

**Q3 Main Strengths:**

The paper provides solutions to an important problem. As the authors mention, idea for the algorithms have been influenced by POMDP known approximation techniques. However, the overall approach is novel as new techniques were added. In principle, the approach even deal with performance guarantees. The authors try hard to evaluate the algorithms with three example domains, although space is limited and the description are indeed short. The authors provide material to ensure reproducibility.

**Q4 Main Weakness:**

The paper is not very specific whether error bounds (and the conditions for the bounds) are actually a problem for the example domains in which the algorithms are applied. Are the respective value functions Lipschitz-continuous, i.e., is it obvious that only state and belief matter? Maybe I just missed the point here.

The paper is quite hard  to read due to a very condensed presentation, but the authors made it possible to understand the main lines of argumentation.

**Q5 Detailed Comments To The Authors:**

Abstract: Unlike Grid-Vi --> Unlike Grid-VI
Section 2.1: A solution of an MDP is called a policy (π): Make clear that a policy returns a distribution over the actions if applied to a state.
Proposition 1: Explain H_t for the sake of completeness
Propsoition 2: Explain that b_t^{s_t, a_t, s_{t+1}} is a vector for all types, and refer the reader to (1) for each type.
Section 4.2: when two different belief points b1 and b2 discretizes to the same point --> discretize?
Theorem 1: SSP, the initial value is admissible,  SSP, and the initial value is admissible,
Section 5.2: How should one determine K for practical problems?
Paragraph “Efficient Interpolation”: One key advantage of using a regular grid is that finding λ is more efficient. Is the determination of λ discussed again in the sequel?

**Q9 Complying With Reviewing Instructions:**

Yes

---

> ### Author Rebuttal · Authors · 2024-04-04
>
> Thank you for your thorough review and positive assessment of our work. We are encouraged that you find our algorithms for OAMDPs/OASSPs to be novel, important, and impactful. Below we address some of the specific points you raised:
>
> Regarding whether the Lipschitz conditions apply to the example problems: You raise a good point about whether the value functions in our experiments are Lipschitz continuous, which is an assumption required for our theoretical guarantees. Two of the problems (MazeWorld and BlocksWorld) used in the paper are Lipschitz OAMDPs, where reward functions are Lipschitz, and since they all use the noisy rational model (Eq. 3), belief transitions are Lipschitz (Proposition 3). With sufficiently small $\gamma$, Proposition 2 guarantees that the value functions are Lipschitz. When \gamma is large and Proposition 2 does not apply, we technically do not provide a guarantee of the Lipschitz-continuity of the value functions. However, empirically speaking, the two problems seem to have Lipschitz-continuous value functions. The other problem (Acronym) uses an entropy-based reward function, which is not Lipschitz. We will clarify this point in the final version.
>
> Regarding the condensed presentation: We agree the paper is quite dense due to space constraints. To improve clarity, we will expand on notation like H_t and b_t in the propositions as you suggest. We'll also fix the typos you spotted.
>
> Regarding how to determine λ: The manuscript currently does not explain how to determine λ in detail and cites Zhou and Hansen (2001) for a reference. While we will not have enough space to explain the algorithm in the main text, we will add the explanation in the appendix in the final version.
>
> Reproducibility: We will expand our descriptions of problem instances in the appendix and provide the source code for our experiments.

---

### Official Review · Reviewer_hNBn · 2024-03-27

**Q2-1 Originality-Novelty:** 2
**Q2-2 Correctness-Technical Quality:** 3
**Q2-5 Clarity Of Writing:** 3

**Q1 Summary And Contributions:**

This paper presents algorithms for solving OAMDPs (Observer Aware MDPs), and provides theoretical results and experimental evaluation of those algorithms.

**Q2-3 Extent To Which Claims Are Supported By Evidence:**

3: Good: the main claims are supported by convincing evidence (in the form of adequate experimental evaluation, proofs, (pseudo-)code, references, assumptions).

**Q2-4 Reproducibility:**

3: Good: key resources (e.g. proofs, code, data) are available and key details (e.g. proofs, experimental setup) are sufficiently well-described for competent researchers to confidently reproduce the main results.

**Q3 Main Strengths:**

This paper presents algorithms for solving OAMDPs (Observer Aware MDPs), and provides theoretical results and experimental evaluation of those algorithms. These all seem new.

OAMDPs is an interesting class of problems in which the domain is fully observed by the actor, but the rewards are provided by an observer with only partial observability of the domain. The class of problems is of interest to the UAI community.

**Q4 Main Weakness:**

The contributions seem to be simple extensions of previous works, drawing on the relatively close relationship between OAMDPs and MDPs and POMDPs. In particular, it is seems these algorithms are simplifications over existing algorithms for POMDPs.

**Q5 Detailed Comments To The Authors:**

The paper seems to make relatively straightforward innovations in algorithm design and theory, so a more thorough and extensive experimentation (with more examples, comparing to more from algorithms for POMDPs and other approaches) and better quantitative (numbers, computational complexity, approximation results) and qualitative (problems belonging to one and not another, and insights into how the differences matter in this case)  could make this paper more valuable to readers.

**Q9 Complying With Reviewing Instructions:**

Yes

---

> ### Author Rebuttal · Authors · 2024-04-04
>
> Thank you for your review of our work. We appreciate your recognition of the problem's relevance to the UAI community. We wish to clarify why our contributions aren’t simple extensions of previous works. Below we address your main concerns and highlight the novelty of our approach.
>
> Comparison with POMDP algorithms: We would like to first emphasize that OAMDPs and POMDPs are two distinct problems with neither of them subsuming each other. In OAMDPs, the belief represents the observer's uncertainty about the acting agent's type, whereas in POMDPs, the belief captures the acting agent's uncertainty about the state of the environment. This distinction leads to fundamentally different belief update dynamics between the two frameworks. In an OAMDP, the agent is fully aware of its true type from the outset, so the observations it receives at each step do not provide any new information. The observer, on the other hand, treats the agent's actions as evidence that incrementally reveals the agent's type over time.
>
> Since OAMDPs are distinct problems from POMDPs, the existing algorithms for POMDPs are not directly applicable to OAMDPs. In particular, OAMDPs do not have piece-wise linear convex value (PWLC) functions, which most algorithms for POMDPs rely on. Prior work on OAMDPs (https://proceedings.mlr.press/v161/miura21a.html) articulates in greater detail the differences between OAMDPs and POMDPs, highlighting the need for specialized solution methods that account for the unique belief update structure in OAMDPs. We will clarify this point in the revision.
>
> Novelty over POMDP and continuous MDP algorithms: We believe that our heuristic search-based variants (Grid-(L)RTDP) are based on new algorithmic insights that exploit the unique structure of OAMDP/SSPs. The key insight is that, in OAMDP/SSPs, physical state transitions and termination conditions only depend on the current physical states and not on the observer’s belief. This property preserves the existence of a proper policy for discretized problems (Lemma 2), and enables guaranteeing convergence (Proposition 6). Note that the same property does not necessarily hold for discretized POMDPs and continuous MDPs. This is because, in these problems, the transition dynamics depend on the discretized states, which might not reflect the original transition dynamics perfectly. The closest POMDP algorithm to Grid-RTDP is RTDP-bel (Bonet and Geffner 2009), which is a heuristic search algorithm for POMDPs based on belief discretization. However, RTDP-bel is known to suffer from value instabilities. Grid-RTDP avoids this with a carefully designed discretization scheme. We will clarify this point in the revision.
>
> More experimental evaluations: While more extensive experiments could strengthen the empirical contributions, we have clarified that POMDP algorithms based on PWLC of value functions are not directly applicable to OAMDPs, and thus direct comparisons are not possible. Other POMDP algorithms not based on the PWLC property exist, the most famous of which is perhaps POMCP (Silver et al., 2010). POMCP is a Monte-Carlo Tree Search algorithm, which our paper performs comparisons against. As suggested by other reviewers, however, we plan to add details about the experiments in the revised version.
>
> Theoretical analysis: We agree that the paper does not describe the computational complexity and relationship with other models in depth. Previous work on OAMDP (https://proceedings.mlr.press/v161/miura21a.html) investigated the theoretical aspects of the model. In short, previously published work showed that computing an optimal policy for OAMDPs is PSPACE-hard and that OAMDPs and POMDPs are distinct problems, but both are special cases of more general multi-agent problems called I-POMDPs. In the revised version, we will mention these theoretical aspects so that the differences from POMDPs are clearer to the reader.
>
> We believe that these clarifications make the novelty and significance of our contributions more apparent. Our specialized algorithms for OAMDPs, while building on insights from prior work, are not simple extensions but rather carefully designed techniques that leverage the unique properties of this important class of problems.

---

### Meta-Review · Area_Chair_kSfG · 2024-04-16

**Summary:** The paper considers a class of MDPs where the reward depends on the MDP state and the belief-state of an observer. This setting cannot be trivially translated into a POMDP, but bears a lot of resemblence and allows to translate many techniques and standard results from that literature. The main contribution of the paper is to present two approximate algorithms for solving OAMDPs: one based on (discretized) value iteration, and one based on (real-time) dynamic programming. The algorithms are derived and theoretically analyzed, and their performance is empirically compared against a previously proposed algorithm (with similar performance but without theoretical guarantees).

**Recommendation:** The paper has received a very broad spread of initial scores. Unfortunately the lowest and highest scoring reviewers have not responded to the rebuttal and have not participated in the reviewer discussion. I personally think that most issues raised by the negative reviewer have been addressed by the authors; but I also believe that the highest positive score of a 9 (technically flawless, very convincing evidence, significant impact) is either a mis-click or inflated, in light of the issues raised by the other reviewers. Taking all information together, I do believe the paper provides an interesting and solid theoretical (approximate) solution and analysis, with some early experimental evaluation that could be expanded to further strengthen the paper. Additionally, the presentation of the paper is good but could be improved to be even more accessible to a broad audience. The fact that no reviewer championed the paper during the AC-reviewer discussion (instead two reviewers pointed out that they still believe the paper is borderline) is quite representative of the work, I think. While I have no technical objections, the question of whether I'd estimate the paper to fall into the top 25% of submissions does not have a very clear 'yes' as an answer. I therefore currently recommend acceptance, noting that it is one of the two the weakest accepts in my batch. Since I do not have visibility over the quality of all submissions (but only a very limited batch) the ultimate decision must be made in relation to the other submissions - I would not object to the paper being rejected (in which case I would want to encourage the authors to use the feedback from the reviewers to produce the most interesting and impactful next version of the manuscript).